# The effect of foot-stretcher position and stroke rate on ergometer rowing kinematics

Ian Engstrom[1], Katelyn Anderson[1], Eleanna Bez[1], Cristine Agresta[2], Scott Telfer [1,3,4] *

1 Department of Orthopaedics and Sports Medicine, UW Medicine, University of Washington, Seattle, WA, United States of America, 2 Department of Rehabilitation Medicine, UW Medicine, University of Washington, Seattle, WA, United States of America, 3 RR&D Center for Limb Loss and MoBility (CLiMB), VA Puget Sound, Seattle, WA, United States of America, 4 Department of Mechanical Engineering, University of Washington, Seattle, WA, United States of America

* telfers@uw.edu

## Abstract

Rowing ergometers are popular tools for general fitness and competitive crew teams. The effect of the equipment set up on the rowing stroke has received limited attention. This study aimed to determine the effects of altering the foot-stretcher position on rowing kinematics across different stroke rates. Eleven college-level rowers took part in this study. A rowing ergometer was modified to allow the height and angle of the foot-stretcher to be adjusted. Seven foot-stretcher positions were tested, each at rates of 22, 26, and 32 strokes per minute. Sagittal plane kinematic waveforms were compared between conditions for all major joints using statistical parametric mapping, and temporal variables were assessed ($p <$ 0.05). Stroke rate was found to affect kinematic patterns for all joints. The effect of the foot-stretcher position was limited to the ankle and hip. Similarly, the timing of events during the rowing stroke was affected by the stroke rate, but not foot position. These results indicate that while some limited changes to the stroke technique can be caused by altering the foot-stretcher position, the changes were largely compensated for by the rowers and are generally smaller than differences between stroke rates.

## Introduction

Rowing is a popular activity, both as a competitive sport and for general fitness, with a 2017 survey estimating that there were 11.7 million rowing ergometer (land-based rowing machines) users in the US alone [1]. The rowing stroke is a complex, whole body motion that requires coordinated movements to produce a smooth and efficient transfer of power to either the boat or the land-based ergometer [2, 3]. Previously, researchers have investigated the relationship between the rower and the setup of the boat, and how that relationship affects performance, power output, and the kinematics of the rowing stroke [4]. While rowing, athletes physically interact with only three components of the boat or ergometer: the handle, seat and foot-stretcher. The foot-stretcher is perhaps the most easily adjustable of these components and a change in its position is often suggested with the aim of affecting the rowing technique [5, 6].

**Data Availability Statement:** The data underlying the results presented in the study are available from https://github.com/Telfer/Rowing_Biomechanics

**Funding:** The author(s) received no specific funding for this work.

**Competing interests:** The authors have declared that no competing interests exist.

Several studies have looked at the biomechanics of ergometer rowing, finding changes in stroke kinematics as a result of factors including the type of ergometer used [7], the stroke rate [8], sex [9], and level of experience of the participants [10]. The effects of foot-stretcher position has been investigated by several authors. Caplan and Gardner tested 10 rowers on indoor ergometers with foot-stretcher positions 50mm and 100mm above the standard position and found that raising the foot helped to maintain mean power per stroke during maximal rowing for 3 minutes 30 seconds [11]. In addition, the authors looked at changes in body position at distinct timepoints during the stroke and found the most consistent effects due to altered foot position to be changes in the shank angle, along with limited differences in the knee, hip, and trunk angles [12]. Similarly, Buckeridge et al found small changes in rowing kinematics during 1 minute race pace tests over 4 different stretcher heights, mainly at the catch position. The effect of changing the angle of the foot-stretcher has been tested, and increasing the angle to 46° was found to increase power output over 2000m tests [13]. On water, raising the foot stretcher height was found to affect several kinematic variables during 200m tests at race pace, showing both positive and negative influences [6].

Stroke rate has been found to influence rowing stroke biomechanics, leading to changes in pelvic rotation [8] and muscle activation patterns [14]. Training programs for rowers generally consist of rowing at different stroke rates, both within and between sessions, with power output and exercise intensity tending to increase with rising stroke rates [8]. While the effect of foot-stretcher position has been studied at race pace intensity, its effect across a range of stroke rates commonly used in training programs has not been assessed.

During a year of training and racing, particularly for those at the collegiate level and higher, the rowing stroke can be repeated hundreds of thousands of times and can apply high levels of stress on the body. It has been noted that the incidence of injury in rowers is high, with the back and ribs being common sites for overuse injuries [15]. Researchers have postulated that the set up of the ergometer or boat may be a modifiable factor for injury risk [12] with for example a high foot-stretcher position thought to be related to an increased prevalence of back injuries [16]. However, no clearly defined mechanism has been presented for this, and no evidence of an association has been published in the literature.

The aim of this study was to determine the effects of altering foot-stretcher position on the angles of the major joints of the body during the rowing stroke across different stroke rates. We hypothesized that changes to the height and angle of the foot-stretcher would lead to significant changes in rowing kinematics, the timing of key events within the stroke, and also the rowers' Rating of Perceived Exertion (RPE). Specifically, we anticipated significant differences in hip and back kinematics as a result of the changes in foot position. In addition, we hypothesized that there would be differences in the rowing kinematics across different stroke rates.

## Materials and methods

### Study design

This was a laboratory-based, single visit, repeated measures study of rowing kinematics. The study was approved in advance by the University of Washington institutional review board (ref: STUDY00007799).

### Participants

Eleven rowers took part in the study between March 2020 and September 2021. Demographic information is provided in Table 1. All participants provided written informed consent upon enrollment into the study. To be eligible for inclusion, potential participants were required to have at least 1 year's rowing experience at a collegiate level, and currently row regularly (at

**Table 1. Participant demographics.**

|  | Age (years) | Height (m) | Leg length* (m) | Trunk to leg length ratio (%) | Mass (kg) |
|---|---|---|---|---|---|
| Female (n = 6) | 21.6 (20.8–22.8) | 1.83 (1.75–1.98) | 1.06 (1.02–1.08) | 68.3 (66.2–69.7) | 72.4 (62.0–80.5) |
| Male (n = 5) | 22.4 (21.0–27.3) | 1.92 (1.85–1.99) | 1.11 (1.05–1.20) | 66.7 (63.4–69.6) | 90.8 (74.8–99.5) |
| All | 22.0 (20.8–27.3) | 1.91 (1.75–1.99) | 1.08 (1.02–1.20) | 67.4 (63.4–69.7) | 82.4 (62.0–99.5) |

All data presented as mean and range.

*Height of the anterior superior iliac spine from ground during relaxed standing

least 3 on water or ergometer sessions of >1 hour per week). They also had to have no current or recent (within 6 weeks) injury that had limited their ability to row, defined as requiring them to miss two or more consecutive days of training. Given the exploratory nature of the study no formal sample size calculation was performed, however we based our recruitment target on those for similar studies in the prevailing literature [7, 11].

## Equipment

A rowing ergometer (Model D; Concept 2 Inc., Morrisville, VT, USA) was modified to allow the overall foot-stretcher position to be adjusted up and down along the plane of the foot-stretcher, and the angle changed (Fig 1). Note that the strap and heel position did not move relative to the foot during these adjustments, and this was set to the rower's preferred position at the start of testing. This setup allowed for seven foot-stretcher positions to be tested: 1) the ergometer's standard position (henceforth referred to as neutral); 2) neutral raised by 20mm (plus 20); 3) neutral raised by 40mm (plus 40); 4) neutral lowered by 20mm (minus 20); 5) neutral lowered by 40mm (minus 40); 6) foot-stretcher tilted from neutral by 5 degrees with toes moving towards body and adjusted to keep the foot strap at the neutral position height (toe up); and 7) foot-stretcher tilted from neutral by 5 degrees, toes away from the body, similarly adjusted for height (toe down). The height and angle are both common changes to the foot-stretcher position that have been reported in the literature [5].

Participants wore their own footwear and tight-fitting apparel for testing. Retroreflective markers were securely attached to joints and segments, allowing kinematic tracking of feet, shanks, thighs, pelvis, lower and upper back, upper arms, forearms, and hands. Specifically, for the lower limb individual markers were attached bilaterally to the dorsal surface of the hallux, the base of metatarsal 5, the lateral side of the calcaneus, lateral and medial malleoli, the lateral and medial epicondyles of the knee, and the anterior superior iliac spine, highest point of the iliac crest, posterior superior iliac spine. On the torso, individual markers were placed on the medial and lateral lower back, T10 vertebra, medial and lateral upper back, C7 vertebra, manubrium, xiphoid process, and acromion processes. On the arms, markers were placed on the medial and lateral epicondyles of the elbow, radial and ulnar styloids, and midpoint of the third metacarpal on the dorsal surface. Clusters of four markers mounted on rigid plates were also attached to the shanks, thighs and upper arms. Further details on the marker model used are provided in the S2 File. The marker model was developed based on preliminary work to minimize marker drop out during the rowing stroke. For example, at the catch, the anterior superior iliac spine markers are generally occluded by the thigh and trunk, therefore additional tracking markers were placed on the posterior of the pelvis to ensure that we could track the segment. An 8-camera optical motion capture system recording at 100Hz was used to track the position of the markers during the testing (Vicon, Inc., Oxford, UK).

Participants were asked to set up the ergometer as they would normally with regards to foot strap position and damper setting. After the set up was complete, and prior to the start of the

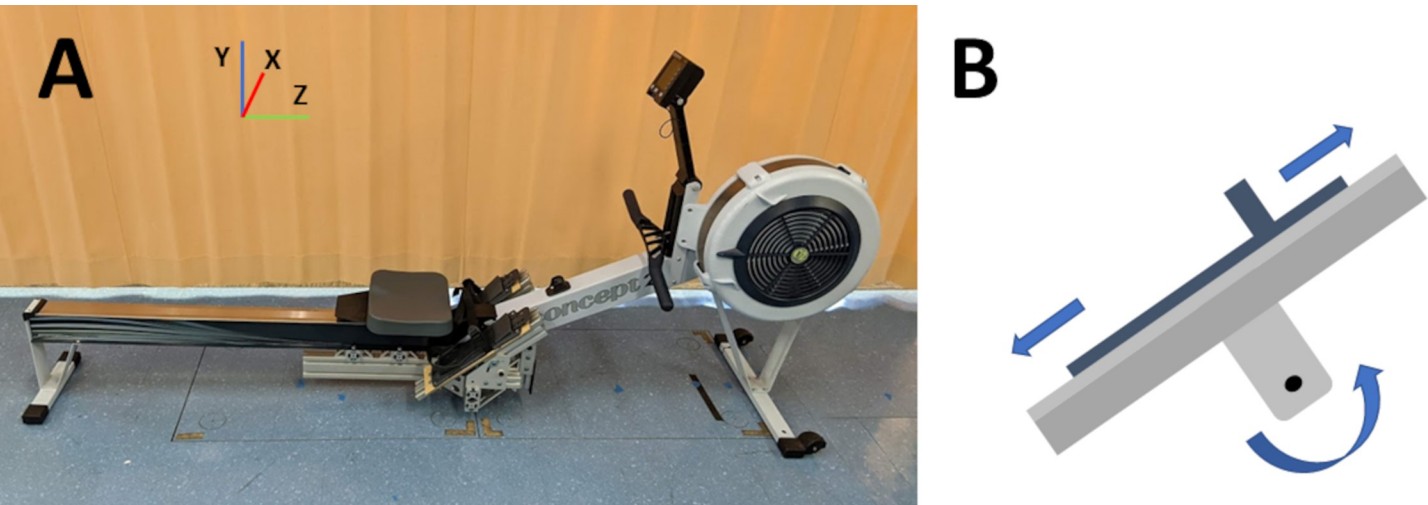

**Fig 1.** A) Customized rowing ergometer with adjustable foot-stretcher allowing position to be raised, lowered, and/or the angle changed. B) Schematic of adjustable foot-stretcher showing adjustments in height and angle.

data collection, participants were given at least 10 minutes to warm up using a self-selected protocol on the ergometer with the foot-stretcher in the neutral position.

## Data collection

At each of the seven foot-stretcher positions, participants were asked to row at 22 strokes per minute (spm) for 60 seconds, followed by 60 seconds rest or light rowing, then row at 26 spm for 60 seconds, followed by 60 seconds rest or light rowing, and finally row at 32 spm for 60 seconds. The order of the stroke rates was not randomized as pilot testing found rowing kinematics to be more consistent within stroke rates when the stroke rate and power output steadily increased. These stroke rates were chosen as they are commonly used in training programs and all participants were familiar with them. The order of testing for the foot-stretcher conditions was randomized in advance. Across all foot-stretcher conditions, participants were asked to maintain a consistent, self-selected power output for each stroke rate, measured as time split per 500m and with a window of +/- 1 second. (For example, a participant could choose to maintain a 2:00 split time every time they rowed at rate 22, 1:55 at rate 26, and 1:50 at rate 32). These power outputs were self-selected so that the rowers could choose levels that they were able to maintain throughout the test period without the effects of fatigue having a significant effect on their technique. Participants had approximately 3 minutes of rest between each condition while the position of the foot-stretcher was changed. After each condition, the participant was asked to give their peak RPE using the Borg Category-Ratio Scale, a numerical instrument that asks for a rating between 0 and 11 [17].

## Data processing

After labeling and exporting the motion capture marker data, data were post-processed in Visual 3D (C-Motion Inc., Germantown, MD, USA). A link segment model using 6 degree-of-freedom joints was defined using the marker data (see S2 File for detailed information on segments). Using this model, joint angles were calculated for the ankle, knee, hip, lumbar sacro-lumbar, thoracolumbar, shoulder, and elbow joints. The proximal segment used as the reference for the joint (i.e., the thigh was used as the reference segment for the knee).

The key events in the rowing stroke cycle were defined as the "catch" and the "finish". The "catch" was defined as the time point when the marker on the participant's hand began moving in the negative Z direction (see S2 File) i.e., away from the ergometer screen at the start of the drive phase. The "finish" was defined as the time point when the handle ceased moving in the negative Z direction. A complete stroke cycle was defined from finish event to subsequent finish event.

Kinematic waveforms describing joint angles, normalized to 101 points for each stroke cycle, were determined for each trial (seven foot positions at 3 stroke rates, giving 21 trials per participant). The middle 10 strokes of each trial were used to determine the mean stroke cycle kinematics as these were found to produce reliably consistent data in terms of stroke rate and overall kinematics. A kinematic waveform describing joint angles was produced for the ankle, knee, hip, shoulder, elbow, sacrolumbar, and thoracolumbar joints. Given that the rowing stroke is primarily limited to single plane joint movements (flexion/extension) we focused on these during this study. As with previous studies [18], due to the symmetric nature of rowing we only analyzed one side of the body (randomly selected) for bilateral joints.

### Statistical analysis

Statistical analysis was performed using R and Python. Figures were produced using the ggplot2 package [19]. Descriptive statistics were calculated for demographic data. The primary analysis of the kinematic data was based on statistical parametric mapping [20], which was used to compare joint angle waveforms across and between conditions. This technique has the advantage of allowing the full motion to be compared, rather than discrete variables within the stroke. A two-way repeated measures analysis of variance was performed for each joint to test for significant effects of stroke rate and foot-stretcher position along with any interactions. This was followed by pairwise comparisons where indicated. The level of statistical significance ($\alpha = 0.05$) was adjusted for multiple comparisons via Bonferroni correction on a per joint basis.

In addition, the effect of foot position and stroke rate on the timing of the rowers reaching the catch position within the cycle, essentially the drive-recovery ratio, was tested for significant effects. After assessing for normality via inspection of Q-Q plots, a two-way repeated measures analysis of variance followed by pairwise comparisons (with Bonferroni adjustment for multiple comparisons) was performed, including 95% confidence intervals. To assess if sex had any effect on the results we also performed a sub-group analysis of this variable. Secondary timing variables, indicating important events during the rowing stroke time to peak knee extension, time difference between peak knee and peak trunk extension were similarly tested. Participant reported scores for RPE across different foot positions were compared using ordinal logistic regression followed by Wilcoxon signed-rank tests for paired comparisons if significant effects were found at the analysis of variance level.

## Results

Please note that given the amount of kinematic data presented and the limitations of the scientific paper format, we have also made the results available online where they can be viewed and explored in more detail: https://telferbiomech.shinyapps.io/rowingfootposition/ (accessed 2023-04-17) and https://github.com/Telfer/Rowing_Biomechanics (accessed 2023-04-17).

### Foot position

Significant differences in joint kinematics due to altered foot position were primarily limited to the ankle, with some small effects seen at the hip (Fig 2). Changes at the ankle were mainly seen during throughout the recovery phase (0–40%) of the stroke and the later part of the

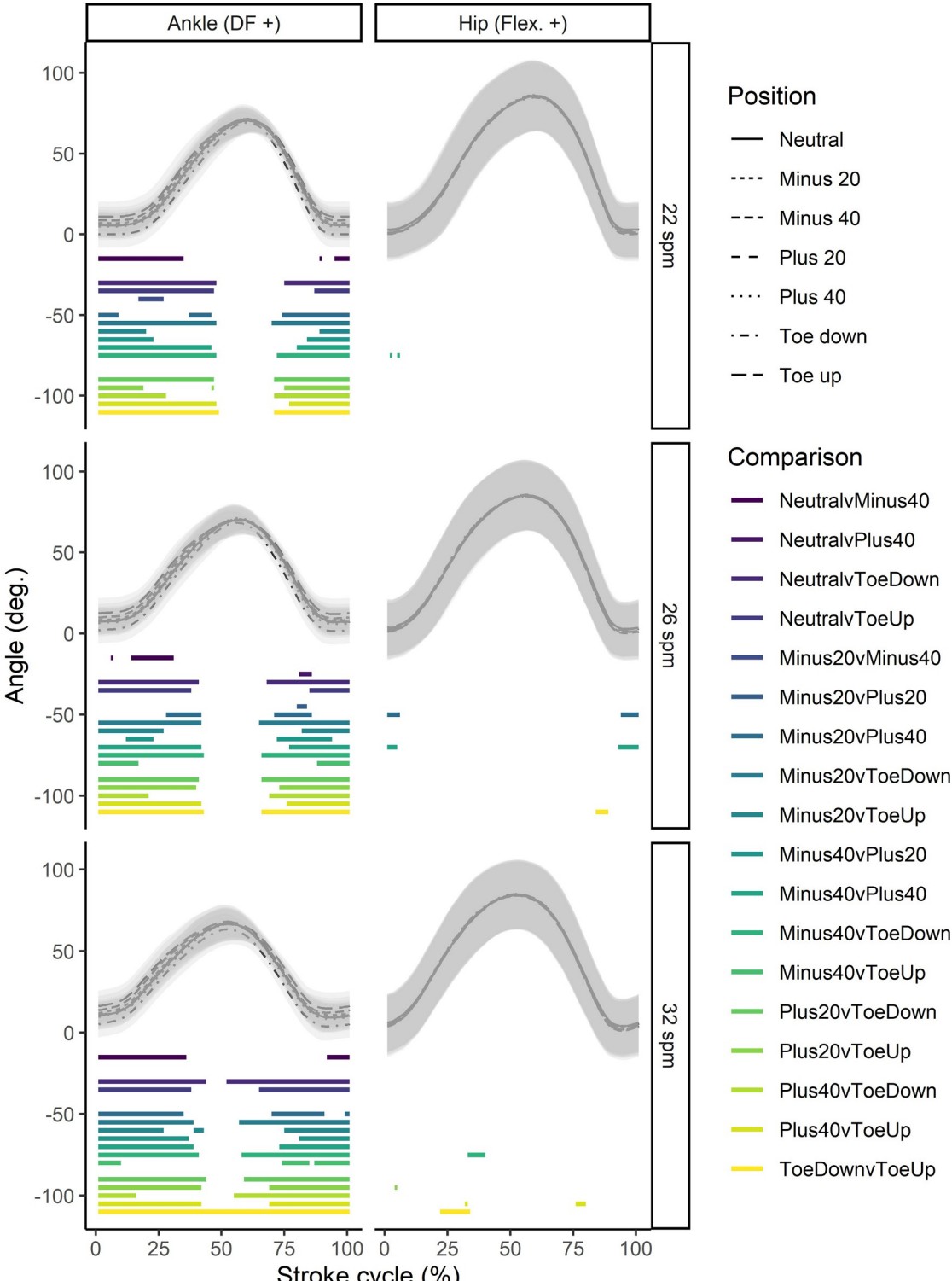

**Fig 2. Foot position comparisons for body joints (columns) where differences were detected across different stroke rates (rows).** Color bars below the waveforms indicate periods of the stroke cycle during which pairwise statistically significant differences were seen. Angles are normalized to the finish position. DF: dorsiflexion; Flex.: flexion; Ext.: extension.

drive phase (75–100% of the stroke). No differences were seen around the catch. These effects were relatively consistent across stroke rates. The most notable changes in the stroke were a result of changes to the foot-stretcher angle (i.e., the Toe up and Toe Down positions). No significant interactions were found between stroke rate and foot-stretcher position.

### Stroke rate

All joint kinematics were found to be significantly different between stroke rates. In general, the joints of the lower limb, including the hip, showed large and consistent differences through large portions of the stroke (Fig 3). The amount of flexion and thus overall range of motion tended to reduce with increasing stroke rate, and the motion curves tended to be shift left, showing earlier joint flexion during the recovery and earlier initiation of extension to begin the drive. Similar patterns could be seen for both the back and upper limb joints (Figs 4 and 5), although statistically significant differences were less consistent than for the lower limb joints. In particular at the higher stroke rate, greater flexion of the thoracolumbar joint could be seen around the catch, and this occurred earlier in the stroke relative to the other rates.

### Stroke timing

The timing of the rowers reaching the catch position during the stroke cycle was found to be significantly influenced by stroke rate ($p < 0.001$, $F_{(2, 20)} = [297.7]$) but not foot position ($p = 0.37$, $F_{(6, 60)} = [1.098]$). There was no significant interaction between foot position and stroke rate ($p = 0.96$). At 22 spm, rowers reached the catch position 60.6% (SD 1.7) of their way through the stroke cycle. At 26 spm, it was 57.0% (SD 1.9) and at 32 spm 53.2% (SD 2.0). All changes in stroke rates resulted in significantly different catch timing ($p < 0.001$, Table 2). Secondary timing variables, time to peak knee extension, time difference between peak knee and peak trunk extension, were similarly affected by stroke rate but not foot position. Similarly, when assessing the effect of sex on catch timing, there was a significant effect seen for stroke rate (female: $p < 0.001$, $F_{(2,10)} = [88.1]$; male: $p < 0.001$, $F_{(2, 8)} = [964.7]$), but not foot position for both groups (female: $p = 0.56$, $F_{(6, 30)} = 0.8$; male: $p = 0.16$, $F_{(6, 24)} [1.7]$).

### Rating of perceived exertion results

No significant differences were found between perceived rating scores for different foot-stretcher positions (Fig 6).

## Discussion

In this study, we performed a detailed analysis of the effects of changing foot-stretcher position on the kinematics of the rowing stroke across different stroke rates. We found significant changes in rowing kinematics at the different stroke rates across all joints. In particular, for the lower limb joints, the motion curves were shifted to the left as stroke rate increased. This was also reflected in the timing of the catch shifting earlier in the stroke cycle as the stroke rate increased. While the absolute time for each phase of the stroke reduced with increased stroking rate and power output, the change in relative catch timing represents a reduced proportion of stroke time spent in the recovery phase of the stroke. This is likely to allow the rower enough time to complete the drive phase. These findings are largely in line with previous studies that found stroke rate can alter rowing kinematics [8, 14].

The effect of changing the foot-stretcher position was limited, both for rowing stroke kinematics and perceived exertion. Kinematic effects were primarily at the ankle joint. The significant influence on the ankle joint makes sense as the foot is in direct contact with the foot-

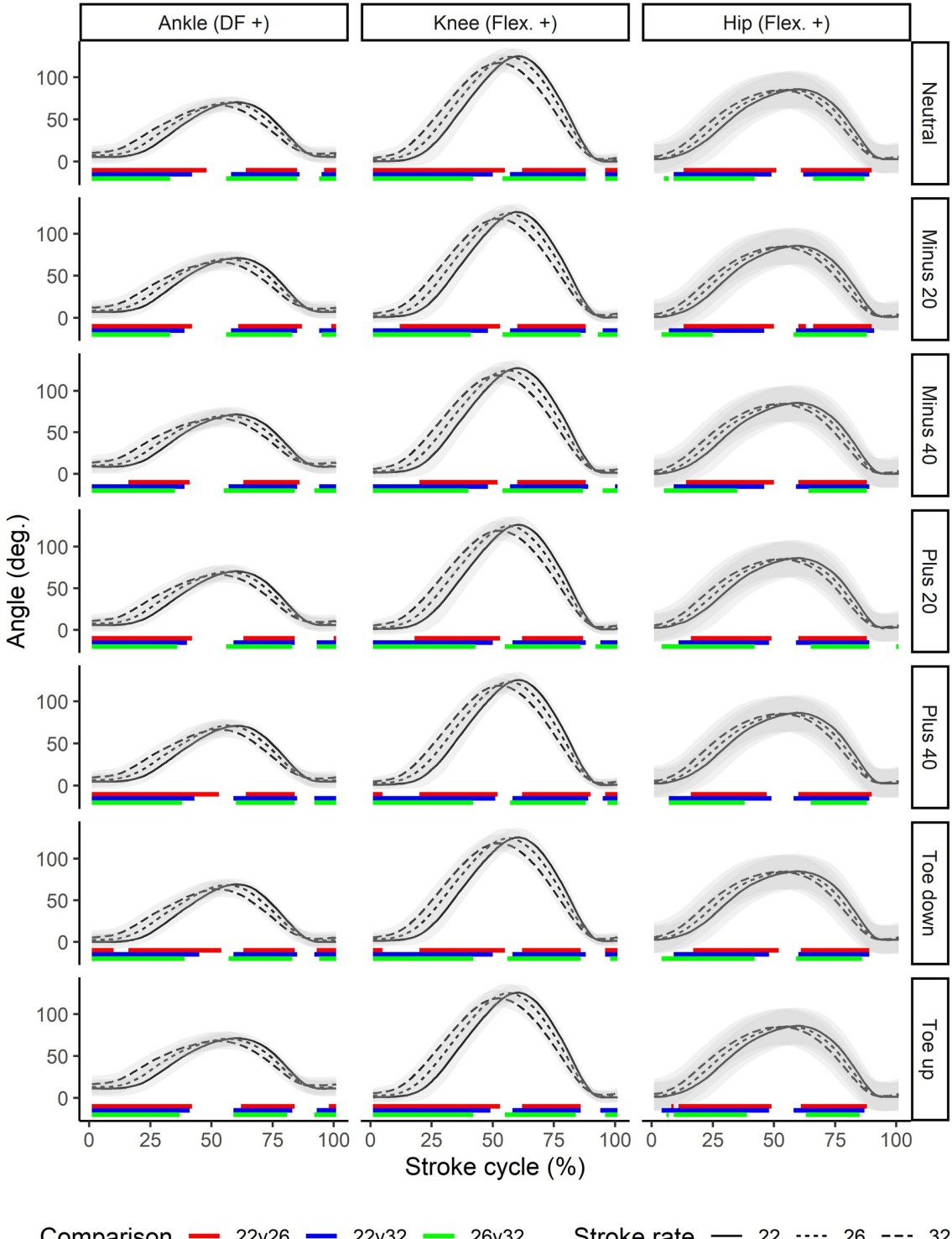

**Fig 3. Stroke rate comparisons for the lower body joints (columns) across different foot positions (rows).** Color bars below the waveforms indicate periods of the stroke cycle during which pairwise statistically significant differences were seen. Angles are normalized to the finish position. DF: dorsiflexion; Flex.: flexion; Ext.: extension.

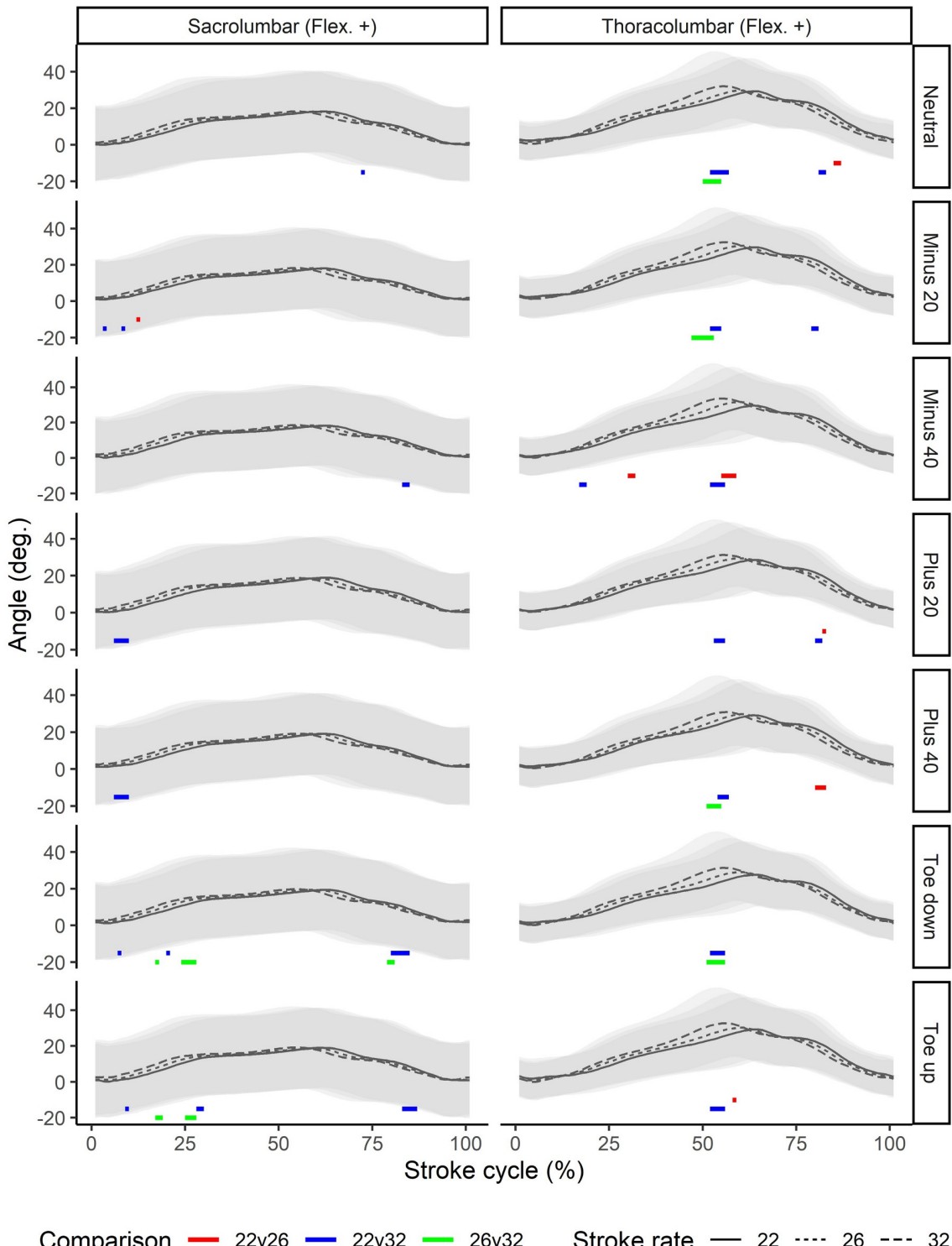

**Fig 4. Stroke rate comparisons for the back joints (columns) across different foot positions (rows).** Color bars below the waveforms indicate periods of the stroke cycle during which pairwise statistically significant differences were seen. Angles are normalized to the finish position. DF: dorsiflexion; Flex.: flexion; Ext.: extension.

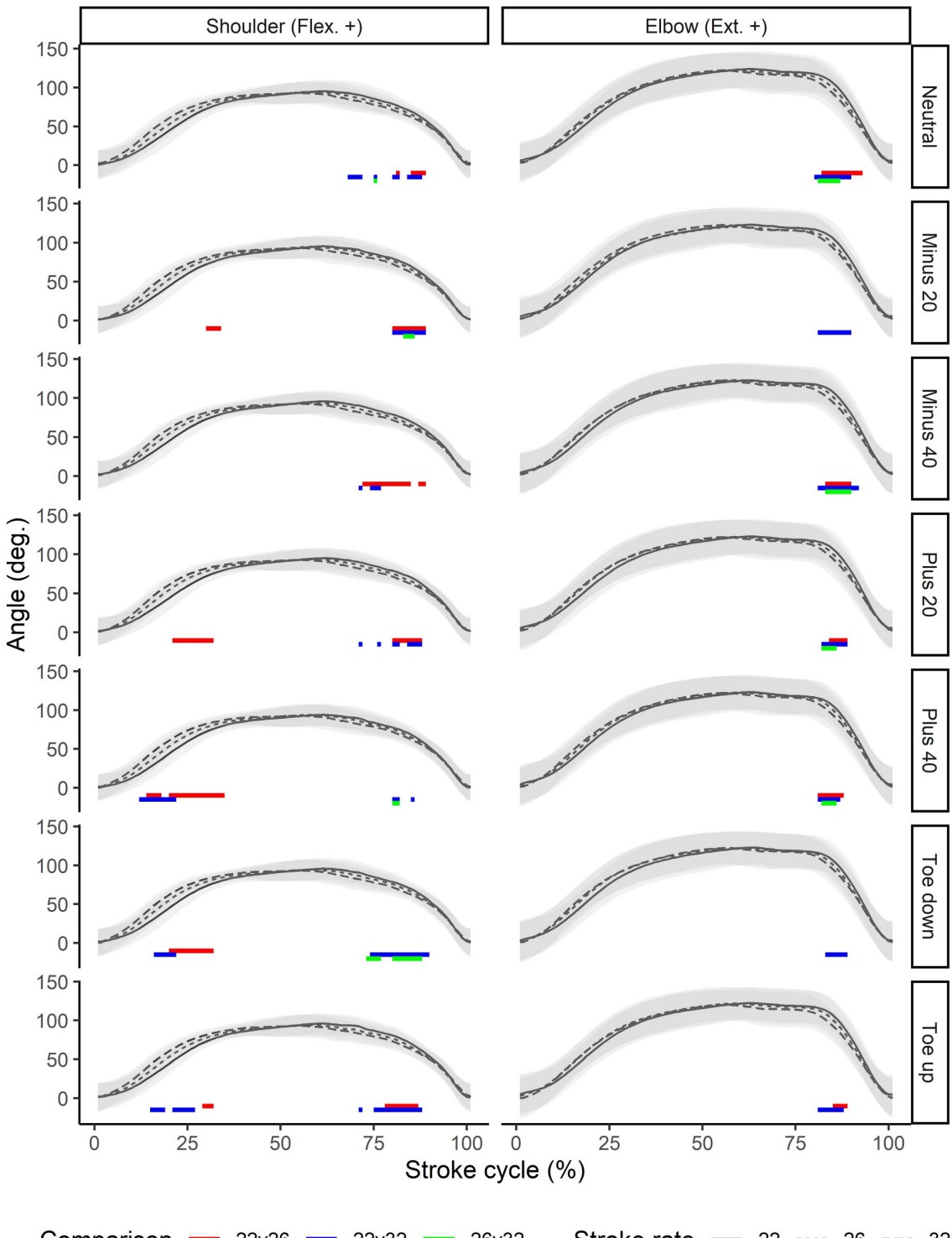

**Fig 5. Stroke rate comparisons for the upper limb joints (columns) across different foot positions (rows).** Color bars below the waveforms indicate periods of the stroke cycle during which pairwise statistically significant differences were seen. Angles are normalized to the finish position. DF: dorsiflexion; Flex.: flexion; Ext.: extension.

**Table 2. Analysis of catch timing by stroke rate.**

| Estimate* | Condition 1 | Condition 2 | 95% CI | Adj. *p* value |
|---|---|---|---|---|
| 3.5 | 22 | 26 | 3.3, 3.7 | <0.001 |
| 7.3 | 22 | 32 | 7.0, 7.6 | <0.001 |
| 3.8 | 26 | 32 | 3.5, 4.0 | <0.001 |

* Estimated change in catch timing as a percentage of rowing stroke. Positive values represent catch occurring earlier in stroke.

stretcher. Given the lack of differences seen at the other joints, it appears that the rowers in the study were largely able to adapt to the change in foot-stretcher position and maintain their regular stroke pattern. Some of the participants in the study did note that, subjectively, the higher foot-stretcher positions felt more uncomfortable, however this did not limit their ability to maintain their stroke rate and power output, and was not associated with changes in the overall RPE ratings. Different rowing styles, such as English orthodox or Fairbairn have been developed across the years [21]. The differences between these styles primarily relate to the relative sequencing and timing of the movements that make up the stroke, and our findings imply that foot-stretcher position is unlikely to affect the rower's ability to adapt to these. Rowing coaches and trainers should be aware of these findings when providing recommendations for ergometer set up.

Previous studies have looked at the effect of changing the foot-stretcher position on rowing biomechanics [12]. Our findings were largely in line with this work, and add to it by showing that the effects are consistent across stroke rates and their related power output changes and at

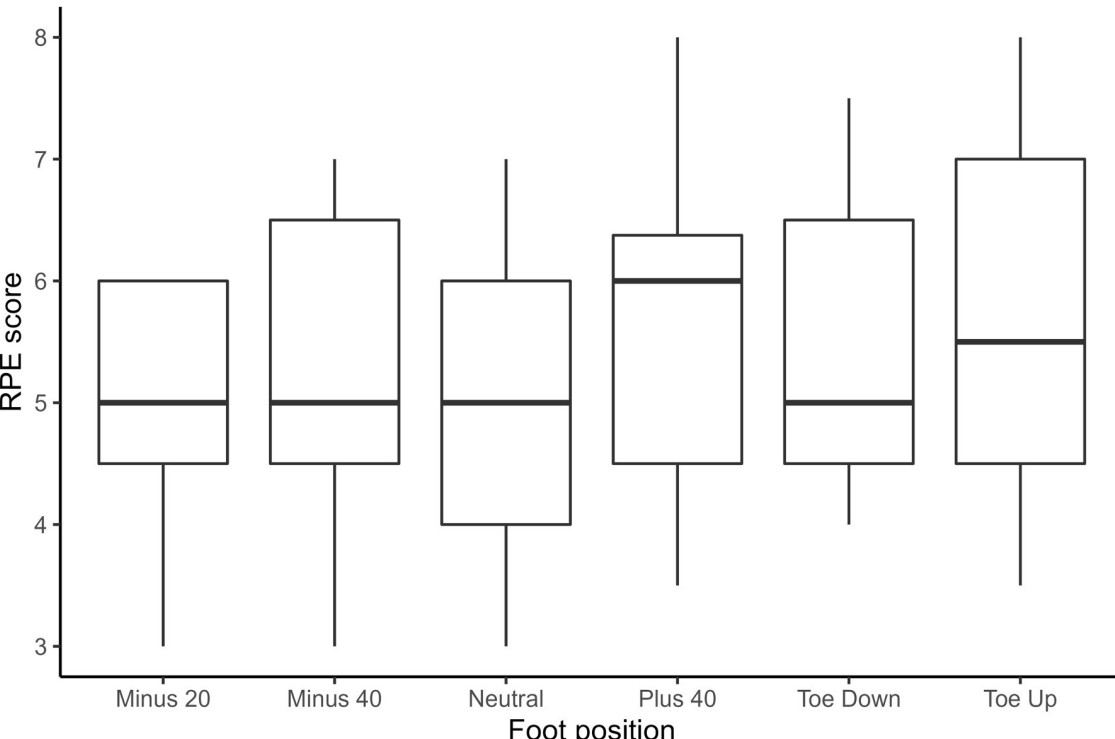

**Fig 6. Boxplots displaying ratings of perceived exertion across foot-stretcher positions displaying median, interquartile range, and maximum and minimum values.**

different foot-stretcher angles. This previous work did note some limited changes in other joints, however this was only found when testing with larger alterations in the height of the foot-stretcher (50mm and 100mm) which we found difficult to achieve on a standard rowing ergometer. Similar to the present study, Buckeridge et al found small but consistent changes in rowing kinematics including at the ankle, hip, and pelvis when increasing the height of the stretcher in 17mm increments up to 51mm above standard. We note that these previous studies tested the rowers at race pace and power outputs. Our study adds to the literature by assessing the changes caused by altering foot-stretcher position across a range of stroke rates that are common in most training programs. In addition, we used statistical parametric mapping to analyze the complete kinematic waveform, providing additional insight into the changes throughout the stroke beyond the assessment of discrete time points or events during the stroke. As we required the participants to maintain the same power output across conditions for each rate (but not between stroke rates), we were not able to assess any direct effects on performance.

Injuries in competitive rowers are common. Spinal loads during ergometer rowing are estimated to be high [22], and lower back and pelvic kinematics during rowing may be associated with injuries [23]. Hypothetically, decreasing the height of the foot-stretcher or tilting the toes forward could reduce stress on the hips and lower back by decreasing the strain on the posterior chain [12]. This may be the case, however we would have expected to see more evidence of a change in back kinematics in our data. Further investigation using a fully instrumented ergometer and musculoskeletal model that would allow individual muscle and joint forces to be estimated thus providing greater insights into spinal loads may be required to investigate this further. Looking at evidence from other sports may provide some insights into these findings. Spinal posture in cyclists with different hamstring extensibility has been studied and it was found that with trunk fully flexed and knees extended, somewhat similar to the rowing position near the end of the leg drive, thoracic and pelvic postures were affected, but not lumbar posture [24]. A study on kayakers found that raising the seat height found some improvements in paddling efficiency, but these were inconsistent between individuals [25].

The study participants were high level athletes; therefore the results may not be applicable to those performing at lower levels. Similarly, differences in rowing biomechanics have been noted between male and female athletes [9] and this study was not designed or powered to test for these. However, visual inspection of the data did not appear to suggest any noticeable differences between these groups.

We note a number of limitations. Changes in power output are also likely to affect rowing kinematics and given that the rowers self-selected their power output at each rate this was not directly controlled for. This was a test of land-based ergometer rowing, and given previously reported similarities in rowing kinematics between land and water training [26], we believe it likely that these results will translate to on-water rowing, however this has not been tested. Given the number of conditions we wanted to test, in order to avoid any fatigue effects the collection period for each condition was by necessity relatively short (this was confirmed by the RPE results), and it is possible that given more time to acclimatize the participants stroke patterns may have evolved. Different types of ergometers are available, particularly those that are dynamic in nature or use different forms of resistance compared to the air-based resistance used in the model tested here. These have previously been found to induce changes in rowing biomechanics [27]. We used skin mounted motion capture markers therefore there are inevitability skin motion artifacts although we attempted to minimize these by avoiding the use of those markers on locations known to be susceptible to high levels of skin movement for tracking tasks. In addition, the analysis focuses on the primary planes of motion for the joints and the rowing stroke involves large ranges of motion so these effects are likely to be minimal.

Finally, our sample was made up of young, collegiate age rowers and included both female and male participants. It is possible that other age groups may vary in their response to foot-stretcher position. No differences were found when performing a subgroup analysis by sex and, given the repeated measures design of the study, it is unlikely that this factor would have a significant effect on the overall results.

This study showed that rowing stroke kinematics can be influenced by relatively small changes in the setup of the ergometer, but that changes of significant magnitude are primarily limited to the ankle joint. Furthermore, any differences resulting from changing the foot-stretcher position are smaller than those caused by altering the power output as a function of increased stroke rate.

## Supporting information

**S1 File. Motion capture model.**
(PPTX)

**S2 File. Animation showing the effect of stroke rate on body sequencing.**
(GIF)

## Author Contributions

**Conceptualization:** Katelyn Anderson, Scott Telfer.

**Data curation:** Ian Engstrom, Eleanna Bez, Scott Telfer.

**Formal analysis:** Cristine Agresta, Scott Telfer.

**Investigation:** Ian Engstrom, Katelyn Anderson, Eleanna Bez, Scott Telfer.

**Methodology:** Ian Engstrom, Katelyn Anderson, Eleanna Bez, Cristine Agresta, Scott Telfer.

**Project administration:** Scott Telfer.

**Software:** Scott Telfer.

**Supervision:** Cristine Agresta, Scott Telfer.

**Validation:** Scott Telfer.

**Visualization:** Scott Telfer.

**Writing – original draft:** Scott Telfer.

**Writing – review & editing:** Ian Engstrom, Katelyn Anderson, Eleanna Bez, Cristine Agresta, Scott Telfer.

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
