## [Decision Letter · Decision Letter 0]

10 Apr 2023

PONE-D-23-06602The Effect of Foot-Stretcher Position and Stroke Rate on Ergometer Rowing KinematicsPLOS ONE

Dear Dr. Telfer,

Thank you for submitting your manuscript to PLOS ONE. After careful consideration, we feel that it has merit but does not fully meet PLOS ONE’s publication criteria as it currently stands. Therefore, we invite you to submit a revised version of the manuscript that addresses the points raised during the review process.

We look forward to receiving your revised manuscript.

Kind regards,

Monika Błaszczyszyn

Academic Editor

PLOS ONE

Journal Requirements:

Reviewers' comments:

Reviewer's Responses to Questions

**Comments to the Author**

1. Is the manuscript technically sound, and do the data support the conclusions?

Reviewer #1: Partly

Reviewer #2: Yes

2. Has the statistical analysis been performed appropriately and rigorously? 

Reviewer #1: No

Reviewer #2: Yes

3. Have the authors made all data underlying the findings in their manuscript fully available?

Reviewer #1: Yes

Reviewer #2: Yes

4. Is the manuscript presented in an intelligible fashion and written in standard English?

Reviewer #1: Yes

Reviewer #2: Yes

5. Review Comments to the Author

Reviewer #1: The work is interesting. In my opinion, technologically advanced. However, from the errors, there is a small study group and not heterogeneous. With such a small number of people, I think this may have affected the results.

• L33 - I would suggest adding statistics on how many people in the world practice rowing.

• L37 – ‘’ (Barrett & Manning, 2004).’’ - Incorrect form of quoting

• Table 1: Participant demographics - Was the number of men and women supported by sample size calculations ?

Also, why did the authors choose this age group ?

• L173 – ‘’ Statistical analysis’’ - I suggest adding an effect size and/or confidence interval.

doi: 10.4300/JGME-D-12-00156.1

• L196 – ‘’ https://telferbiomech.shinyapps.io/rowingfootposition/ ‘’ - this link is not working. You should also add an access date to both links.

• RESULTS - In addition to the graphical form of the presentation of the results, he suggests adding tables with statistical results, this will improve the understanding of the work.

• L32 – ‘’ We note a number of limitations’’ - The authors forget to mention the limitations of heterogeneity. Why did the authors choose to study men and women without differentiating this in the later results? There are differences in the biomechanics of men and women. In my opinion, with such a small group of subjects, this may have affected the results.

doi: 10.1038/s41598-020-76674-2

doi: 10.1016/j.gaitpost.2012.04.006

PMID: 26101903

Reviewer #2: The paper presents the study of determining the effects of altering the foot-stretcher position on rowing kinematics across different 20 stroke rates. The data were collected using the Vicon motion capture system. The materials and method part is well written. I suggest to describe the post-processing step in more detail. During motion capturing some markers may disappear. How was the missing trajectory part solved?

Please explain if the participants had a warm-up before rowing?

6. PLOS authors have the option to publish the peer review history of their article (what does this mean?). If published, this will include your full peer review and any attached files.

Reviewer #1: No

Reviewer #2: No

---

## [Decision Letter · Decision Letter 1]

28 Apr 2023

The Effect of Foot-Stretcher Position and Stroke Rate on Ergometer Rowing Kinematics

PONE-D-23-06602R1

Dear Dr. Telfer,

We’re pleased to inform you that your manuscript has been judged scientifically suitable for publication and will be formally accepted for publication once it meets all outstanding technical requirements.

Kind regards,

Monika Błaszczyszyn

Academic Editor

PLOS ONE

Additional Editor Comments (optional):

Reviewers' comments:

Reviewer's Responses to Questions

**Comments to the Author**

1. If the authors have adequately addressed your comments raised in a previous round of review and you feel that this manuscript is now acceptable for publication, you may indicate that here to bypass the “Comments to the Author” section, enter your conflict of interest statement in the “Confidential to Editor” section, and submit your "Accept" recommendation.

Reviewer #1: All comments have been addressed

2. Is the manuscript technically sound, and do the data support the conclusions?

Reviewer #1: Yes

3. Has the statistical analysis been performed appropriately and rigorously? 

Reviewer #1: Yes

4. Have the authors made all data underlying the findings in their manuscript fully available?

Reviewer #1: (No Response)

5. Is the manuscript presented in an intelligible fashion and written in standard English?

Reviewer #1: Yes

6. Review Comments to the Author

Reviewer #1: (No Response)

7. PLOS authors have the option to publish the peer review history of their article (what does this mean?). If published, this will include your full peer review and any attached files.

Reviewer #1: No

---

## [Editor Report · Acceptance letter]

3 May 2023

PONE-D-23-06602R1 

The Effect of Foot-Stretcher Position and Stroke Rate on Ergometer Rowing Kinematics 

Dear Dr. Telfer:

I'm pleased to inform you that your manuscript has been deemed suitable for publication in PLOS ONE. Congratulations! Your manuscript is now with our production department. 

Kind regards, 

on behalf of

Dr. Monika Błaszczyszyn 

Academic Editor

PLOS ONE